# Arab Youth Involvement in Delinquent Behaviors: Exploring Hirschi's Social Bond Theory from a Qualitative Perspective

Mona Khoury-Kassabri [1,*], Edith Blit-Cohen [1], Mimi Ajzenstadt [1,3] and Lana J. Jeries-Loulou [1,2,*]

1   School of Social Work and Social Welfare, The Hebrew University of Jerusalem, Jerusalem 9190401, Israel;
    edith@blitcohen.com (E.B.-C.); mimi.ajzenstadt@gmail.com (M.A.)
2   Institute of Criminology, University of Cambridge, Cambridge CB2 1TN, UK
3   The Open University of Israel, Ra'anana 4353701, Israel
*   Correspondence: monakh17@gmail.com (M.K.-K.); lana.jeries@mail.huji.ac.il (L.J.J.-L.)

**Abstract:** The current study is based on Travis Hirschi's social bond theory, arguing that the debut of delinquent behavior is a result of the weakening of the ties that bind people to society. According to Hirschi's social bond theory, all youth have the potential to commit delinquent behaviors, but they are kept under control by their relationships with friends, parents, neighbors, teachers, and employers. These factors could serve as protective factors from involvement in crimes. Most studies of social bond theory have been conducted in Western countries among male adolescents using quantitative methods, including cross-cultural approaches. However, the ability of social bond theory to explain youth involvement in delinquency in non-Western cultures, especially conservative and authoritarian cultures such as the Arab one, has not been widely examined. This study aims to fill this gap by exploring youth involvement in delinquency using qualitative research on Palestinian-Arab youth in Israel. Addressing youth delinquency within the unique sociocultural, economic, and political situation of the Arab minority in Israel will help us examine the theory's applicability to the explanation of deviant behavior in a variety of contexts.

**Keywords:** youth delinquency; sociocultural; Arab minority in Israel; qualitative research

## 1. Palestinian-Arab Youth in Israel and Their Involvement in Violence

Studies in Israel have shown that Arab minority youth are disproportionately involved in violence. Arab adolescents (12–17 years old) are involved in 54% of violent crime

Cases, compared with the remaining 46% cases related to Jewish adolescent [1]. This percentage is significantly disproportionate to Arab youth's representation in the general population (27%). The findings of the National Authority for Measurement and Evaluation in Education [1] show that the percentage of Arab students who reported having perpetrated serious physical violence in school was much higher than that of Jewish students (15% vs. 7%, respectively).

The Palestinian-Arabs in Israel comprise 20% of the total Israeli population. Approximately 84% of the Arab population in Israel is Muslim and a smaller percentage of its members are Christian and Druze [2]. This community is characterized by high rates of poverty and unemployment, suffering from low governmental expenditure of public funds for health and social services [3].

Inequality between the Arab and Jewish populations in Israel is also found in the educational system. Arab schools tend to have larger class sizes and fewer teachers [4]. Although over the years there has been an improvement in Arab students' educational achievement, there is still a significant gap between them and Jewish students [5].

The Palestinians in Israel remain a discriminated minority since the 1948 war, even though they are formally recognized as Israeli citizens [6]. Furthermore, it is difficult for Palestinian citizens to feel at home in Israeli society because the State of Israel identifies

with Jewish identity and history, and the country's narratives and symbols are grounded largely in the Jewish majority perspective [7].

The Arab minority is largely characterized by traditional and patriarchal family values [8]. These values emphasize the preservation of cultural and historical roots, the provision of needs through the extended family, a preference for collectivistic over individualistic behaviors, the right of the family to intervene in personal issues, and patriarchal values regarding gender [9–11]. In recent decades, however, the Palestinian-Arab community has undergone several modernizing economic, political, and social changes [4]. Modernization processes have led to changes in traditional family patterns, and a tendency toward a more Western, individualistic lifestyle [12,13]. These changes are manifested in strengthening the status of the nuclear family and weakening the extended family [14]. However, many of today's Israeli Arab nuclear families contain both modern and traditional elements, with the latter as dominant [15,16]. Women's status has also changed: there is a noticeable tendency to adopt a broader ideology regarding women's roles, through access to education and the workforce [17]. Despite these changes, however, only 30% of Arab women in Israel participate in the workforce.

The rapid modernization process in Palestinian-Arab society affects parent-child relationships. Some adolescents have adapted faster than their parents and are more willing to adopt Western values. For example, in Arab society, female adolescents and young women are tightly monitored, and they are expected to comply with social values and norms, especially with respect to relationships with men, modesty, housekeeping skills, and fertility [16,18]. This has resulted in a generation gap [19], which can lead to parent-child conflict, lack of supervision, and reduced harmony [20]. These factors can place children at higher risk for involvement in violence [21].

## 2. Social Bond Theory from a Cross-National Cultural Perspective

As argued by Hirschi [22], social bonds have four elements: attachment, commitment, involvement, and beliefs. In this paper we examine each element in turn, in a wider contextual perspective.

## 3. Attachment

Attachment refers to the affective ties that youth have toward parents, peers, teachers, and others, although their relationship with their parents is the most important [22]. According to the theory, a breakdown of attachment in the life of the child and/or lower levels of parental involvement may increase the likelihood of children's involvement in delinquency and antisocial behavior [22]. Peterson et al. [23] explored the applicability of social bond theory to youth delinquency in South Korea. They found that delinquent behavior among Korean adolescents was found to be more constrained by particular social bonds, such as parental supervision. They explained their results from a cultural standpoint, indicating that in South Korea parents direct greater efforts toward the monitoring aspect in attachment than toward other aspects, such as openness. Their results revealed that this aspect was more effective in disciplining children's misbehavior than in Western cultures.

Despite Peterson et al.'s [23] study, little is known about the association between parental attachment and delinquency in traditional societies in general and in Arab society in particular. The few studies that have explored the association between parental attachment and delinquency among Arab adolescents [24,25] found that attachment to parents had a significant impact on reducing delinquent behaviors.

In Western cultures, there have been extensive studies of the effect of parental attachment on youth behaviors [26,27]. However, many of these studies considered both parents as one entity without exploring the unique effects of mothers and of fathers [24]. Even in studies in which attachment to mothers and fathers was explored separately, there has been a tendency to combine them into one parental measure. In the context of Arab culture, this limitation is critical. The Arab family is characterized by patriarchal family values with male dominance; therefore, treating parents as equal in their effect on children's behavior

might result in findings that do not take into consideration the unique cultural values of the group.

In addition to attachment to parents, the role the peer group plays in an adolescent's life has received much research attention. During the teenage years, young people tend to spend more time with their peer group and less time with their parents. Thus, the peer group takes on a greater role in adolescents' lives [28]. According to social bond theory, attachment to prosocial peers as a social bond element is expected to restrain youth from engaging in deviant behavior, such as committing crime [23]. Conversely, attachment to delinquent peers is correlated with increased delinquency [29,30].

In traditional societies, male gender role values and beliefs may stress the importance of masculine honor, which results in youths' feeling the need to protect their manhood and honor by defending their friends and acting violently to help them. Therefore, the cultural context should be taken into consideration when examining the effects of peers on adolescent behavior. This is highlighted in "street code" theory [31], which explains the high rates of violence among African-Americans in the US by factors such as joblessness, alienation, structural disadvantage, and Black status convergence. Nowacki [31] used longitudinal data from the National Youth Survey to examine the influence of family attachment and adoption of the street code among youth, and whether it differs by gender, and found that family attachment reduced the acceptance of street code for both boys and girls.

## 4. Commitment

Another element of social bond theory is commitment: committed individuals invest time and effort in conventional directions such as obtaining an education. According to this theory, youth with strong commitment are less likely to be involved in acts that will jeopardize their hard-won position [22,32]. However, this perspective overlooks contextual factors that may affect the ability of youth to commit to a school, such as socioeconomic factors affecting the schools' capacity to help children commit and succeed in their studies.

As mentioned, there is significant inequality in the Israeli educational system, both formal and informal [30,33]. The financial support for Arab educational institutions is lower and consequently, Arab schools are equipped with fewer facilities (e.g., libraries, computers, and laboratories) and their classes include more students than their Jewish counterparts [33].

Finally, studies in Israel have shown that Arab students experience significantly high levels of physical and emotional-verbal violence from their teachers, which was shown to be associated with the socioeconomic status of the schools and the students' families [34–36]. Therefore, youth commitment might not only be motivated by individual efforts, but is probably a combination of individual and contextual factors.

## 5. Involvement

This element of Hirschi's [22] theory refers to the fact that a person only has a limited amount of time in the day or in one's life to do activities. Each activity takes up a certain amount of a person's limited time and, with the election to do one activity, a person has less of an opportunity to do something else. The thought behind this premise is that a person may simply be too busy involving himself in conventional activities to find the time to engage in deviant behavior. This idea can be used to explain juvenile delinquency, especially because adolescents are more likely to have leisure time. This gives them more free time and energy to be spent on potentially deviant activities.

As with commitment, Hirschi's theory does not explore socioeconomic or political aspects of the element of involvement. Rabinowitz [37] describes the complex situation of Arabs in Israel in terms of their being a "trapped minority" who experience inequality and discrimination in a wide variety of areas, including education, employment, culture, and politics. The Palestinian Arabs in Israel are officially part of society, but structurally they have been pushed into weakened enclaves through a policy of dispossession, dis-

crimination, and exclusion [38]. This policy has led to the creation of poor neighborhoods with insufficient community services that allow for more unstructured leisure time for adolescents. A report examining youth involvement in activities in their leisure time, showed significant differences between Israeli Jewish and Arab youth in their involvement in after-school activities. While 30% of the Jewish youth participated in youth organizations, only 12% of the Arab youth participated in youth organizations. In addition, 28% of the Jewish youth engaged in sports activities while only 16% of the Arab youth engaged in sports [39]. This shows that for the Jewish youth, the availability of community services means more structured time, leaving less time to engage in delinquency, which is not the case among Arab youth.

## 6. Belief

The last element of Hirschi's [22] theory is belief—the extent to which the individual is committed to the rules of a society. According to the theory, if a person truly believes in and accepts the rules and values of the system he belongs to, then he will not be motivated to be involved in delinquent behavior or to violate these rules. Youth involvement in delinquent behavior may result from a lack of identification with the society's formal legal system and its moral validity [40]. Previous studies have used respect for police and other formal agents that impart society's laws and norms as indicators for the belief element in Hirschi's theory [41,42].

Hasisi [43] indicated that many minority groups perceive law enforcement agencies in general, and the police in particular, in a negative manner. This perspective is shaped by a long-term experience of many minority groups receiving harsher treatment by the police, including in some cases greater use of violence and higher arrest rates than majority groups. As a result, some minority groups believe that the law alienates and discriminates against them, and even creates criminalization of their group [44,45].

A previous study showed that more Arabs than Jews in Israel believed the police violated civil rights. About half of the Jewish respondents expressed confidence in the police compared with a third of Arab respondents. Approximately 80% of the Jewish respondents reported that they would obey the police even if they were dissatisfied with the manner in which they were treated, compared to 57% of the Arab respondents [46]. According to Hasisi and Weitzer [47], lack of trust and belief in legal institutions is based on a strong perception of discrimination by the state and its agencies toward the Arab population. Studies show that when facing difficulties, the Arab population tends to refer to the informal system of the family for help rather than to formal systems [48,49].

## 7. Research Questions

The present study examines the following questions: (1) To what extent are the elements of Hirschi's social bond theory manifested among Arab delinquent boys in Israel? (2) To what extent are these elements influenced by the values and norms of the traditional society to which the boys belong, and by the socioeconomic and political disadvantages they experience in Israeli society?

## 8. Methodology

This qualitative study is informed by in-depth, face-to-face, semi-structured interviews with 15 Muslim Arab youth aged 16–19 ($M = 16.76$, $SD = 1.083$), all citizens living in northern Israel. At the time of interview, most (75%) of the participants did not attend school or had dropped out of school; 25% were still studying or had graduated high school. One fifth (20%) dropped out in ninth grade (approximately at age 14). Less than half of the participants (40%) were employed. Most (69%) of the participants' parents were married, and 23% divorced. Furthermore, 41% and 50% of the participants reported that their fathers and mothers had completed high school, respectively. Most (90%) reported that their family's socioeconomic status was medium–very low; 64.3% reported that their fathers

were employed, while 66.7% reported that their mothers were not. Finally, 80% reported that they were not religious, and 70% reported that their families were not religious.

The boys had committed minor crimes and some of them were supervised by probation officers, who in Israel are social workers. We used a snowball sampling method in order to locate the youth [50]. We employed an Arab graduate psychology student to conduct the interviews that took place at a place that the youth chose and was approved by the interviewer.

After obtaining the participants' consent, the interviews were conducted in Arabic and lasted 60–90 min. All interviews were recorded and transcribed with the participants' agreement, and were subsequently translated into Hebrew. This was done because not all the researchers spoke Arabic, and the translation enabled them to conduct peer debriefing to ensure trustworthiness.

The interview was divided into two parts. The first was a short sociodemographic questionnaire, and the second part was the qualitative interview. The participants were asked about their experience at school, about their neighborhood, their free time and their relationship with family and friends.

### 9. Data Analysis

The data were analyzed deductively: we applied the theory to the data in order to test it. The coding was based on concepts drawn from Hirschi's theory [51]. In the first stage, the researchers read the interviews several times to become empathically acquainted with the interviewees' narratives [52]. Next, units of meaning were anchored around the four elements of social bond theory. In the third stage, we created the categories of interest based on the research questions and then sorted the data into those categories. There were no coding discrepancies between the researchers.

### 10. Reliability and Validity

Reliability was achieved in two ways. First, the researchers employed peer debriefing between themselves. Second, the researchers kept an audit trail throughout the data analysis process that clearly described the steps they took, enabling them to follow their research procedures consistently [53]. It is important to highlight the limitations of this analysis, which examines only claims made by the interviewees, representing their perspectives alone. Notwithstanding this limitation, this study offers a unique opportunity to examine Hirschi's theory within a wide social-cultural context.

### 11. Ethical Considerations

The participants were assured that their anonymity and confidentiality would be maintained. In addition, they signed an informed consent form. All the participants' names were changed and no personal details were revealed. An approval for the interviews was granted by the ethics committee of the authors' institution.

### 12. Findings

The interview findings are anchored around the four elements of social bond theory. Below, we discuss Arab youth involvement in delinquency in light of the unique characteristics of this population.

### 13. Attachment

Most of the interviewed boys (more than 70%) did not find their fathers to be supportive and helpful, for a variety of reasons related to their cultural differences. As Hussain said, Arab fathers spent most of their time at work: "My father comes home very tired and doesn't have energy to talk with us. Most of the time, he isn't available to us". Saleem described his relationship with his father in similar terms: "My dad's not around, as he's working most of the day. I hardly see him and when I do, he's always irritable and violent towards me".



The interviewees reported that it was almost impossible for them to turn to their fathers for support and to share their difficulties with them, as this would be seen as weakness. Because of this perception, these boys tended not to seek their fathers' support. Moreover, more than a quarter of the boys described extreme-risk family situations, where the fathers were violent and abusive toward them and their mothers. They even expressed disgust and hatred towards their fathers:

My father is the most disgusting man in the world. He is just evil and violent. He left home and since then I've had nothing to do with him. I don't know what he does in his life and I don't care. My mother told me he was working somewhere. I just don't want to be in touch with him at all. I was so sick of him . . . he threw me and my brother into the street. He was just crazy. (Asbat)

In contrast to these negative reactions to the fathers, the mothers were described by about half of the interviewed boys as sources of support, usually available to them at home: "My mother doesn't work so she's at home all the time. She listens to me and supports me. She is a good woman, she raised us" (Yusuf).

Most of the boys (more than 70%) loved their mothers and, in cases of abusive relationships, worried about them:

My parents are divorced. I live with my mother . . . I've always witnessed a lot of screaming and arguments between my parents, and it made me run away a lot from home. [I would never want] to hurt my mother. The truth is that when I'm sad I turn to her. (Saleem)

My mother did everything she could to get custody of us. Poor mother: I don't know how she was able to stand [my father]. Once she called the police to take him out of the house because he screamed and was extremely noisy and violent toward her. (Nidal)

Moreover, as presented in more than a quarter of the interviews, the mothers' victimhood when present in the daily experience at home weakened their status in the family, making the boys perceive them as powerless and causing them to question their ability to be considered as a source of help and support.

My mother loves me a lot and talks to me a lot. I love her too, but I also think she's weak because she gives in to Dad. Many times when I got into fights and arguments at school or outside, she stood by me and supported me, unlike my father who treated me with severe violence and beat me. But she couldn't help me. (Hussain)

Another important aspect of attachment is the relationships young people have with significant others outside the family. Most (70%) of the boys described a positive relationship with one or two teachers, mostly male. These teachers seemed to provide the youth with some of the qualities they missed in their traditional Arab fathers. They perceived them as dominant figures who are at the same time loving, protective, and caring. Raafat described having a positive relationship with his teacher:

I loved physical education, because I had a teacher who I really liked. He taught me how to play soccer, which I still play. I remember very well the educational counselor and the physical education teacher because they treated me so well. I told them about my frustrations at school, especially with my English teacher, who I didn't like.

The other significant positive relationship the youth developed was with female probation officers. The boys seemed to perceive them as caring women but, in contrast to their mothers, having the ability and power to help them and make changes in their lives:

Today I'm under the supervision of a probation officer and she is very nice. She helps me a lot: she's the one who encouraged me to find a job and start earning money, and she asked me to continue my matriculation exams so that I'll succeed

at something in life. I thank her for her dedication. She didn't give up on me where others did. She understood that I really want to be a better and less violent person. I go to her once a week and it really makes me feel good. (Faaiz)

Another significant point discussed in social bond theory literature is the association of youth with peers and their effect on them. The boys described strong relationships with peers. They felt strongly protective of their friends, even to the point of fighting on their behalf. Therefore, when they perceive their peers as being under threat, they might act protectively and perhaps sometimes aggressively, according to the cultural expectations of them, as described by Nidal:

My relationships with my friends are very good, we protect each other and support each other. We have a lot in common because we've been together for many years and we've made an agreement that we will be together to protect each other, no matter what ... My friends and I beat the kids who harassed us during a game because one of the guys there cursed at my friend and it made me really angry and I couldn't remain passive about that because friends are supposed to protect each other from everything that happens.

## 14. Commitment

Most (70%) of the boys reported that their schools had a poor curriculum, that they lacked attention from their teachers, and that their classrooms were overcrowded. The boys talked about their difficulties at school which began soon after their entry into the school system. They emphasized how their inability to study and their emotional and behavioral difficulties (such as having ADHD) placed them in a problematic position at the school. Almost all the boys dropped out of school or were moved from one school to another:

I have a problem with attention and concentration and I am known as hyperactive. I couldn't sit still at all. I always had to stand up but the teachers didn't care and didn't take into consideration the problem that I have, either in elementary school or in high school. (Kabir)

## 15. Involvement

Most (80%) of the boys indicated that they were not involved in any after-school or leisure activities. However, many of the youth expressed a desire to participate in extracurricular activities. They claimed that they were unable to participate mainly because there were almost no activities in their neighborhood. The boys show that even when there were a few activities in the neighborhood or in other areas in the city, they could not participate because their families were not able to afford to pay for their participation:

In my free time I play ball a lot and go out with friends. We sit outside in the neighborhood talking and smoking; we go online, Facebook a lot, talk to girls, go to the beach or the mall, watch TV ... I asked my parents to sign me up at a gym, but they said I should work to save up for it and that they couldn't pay because we're in a bad financial situation. (Fahim)

The participants often found themselves idle and disorganized. Without access to recreational activities, the boys kept themselves busy by walking bored around the neighborhood, smoking, and disturbing the neighbors. In this way the lack of involvement in normative behavior resulted in violent and antisocial behavior.

## 16. Belief

Many (70%) of the youth expressed their lack of identification with and trust in legal institutions, especially the police. They seemed to perceive the police as having power over them and not as officers whose aim is to help and protect them and their families. Kabir described the relationship between him and the court:

> I sat in court. They asked me a lot of questions but I didn't answer. I didn't know what they wanted from me . . . The interrogator began to accuse me of stealing a bicycle from one of the houses in the neighborhood. I denied it, of course, but what does it matter what I say? If the court decides to blame me, they will blame me no matter what I say. I don't trust them. It's a hooked game. They see I'm an Arab and that's it. To this day I try to erase this case and cannot. I hate judges and courts. What can I do?

The following argument may best explain the distaste Arab youths have for these institutions:

> I love my friends here, but you feel as though the police and the government are encouraging Arabs to fight so they get in trouble. Look at what is happening in Umm al-Fahm [an Arab town], full of violence and no one stops it because it is in the government's interest to have fewer Arabs here in the country. And this is what's happening in other cities. It's not in their interest for us to have social services and investment in our neighborhoods. They benefit from this situation. They want us to remain poor and have more poverty and problems and criminality. (Jamal)

Based on these perceptions, Arab youth involved in delinquency appear not to believe in the institutions that are responsible for representing and maintaining society's rules and norms. According to them, these institutions represent the hegemony, which wants to oppress the Arab minority. Their perception of these institutions causes some Arab youth not to believe in the norms and to take part in delinquent actions.

## 17. Discussion

The study shows that deep understanding of the cultural, economic, and political context of youth involvement in antisocial behaviors is important for the various elements described in Hirschi's [22] social bond theory and can contribute to its generalizability and cross-cultural understanding. Most studies of social bond theory have been conducted in Western countries among male adolescents using quantitative methods [54–56], including cross-cultural approaches [42,57,58]. Moreover, the studies conducted among conservative cultures [24,59,60] did not address all the dimensions of the Hirschi's theory. This research sought to go beyond these previous works by using a qualitative method, and by examining the applicability of all four component of the social bond theory to explain youth involvement in delinquency in non-Western cultures, specifically conservative and authoritarian cultures such as the Arab one. It did so by testing its relevance and applicability to Arab youth in Israel, in the context of their special position.

Regarding the theory's first element, taking into consideration the socioeconomic and cultural context while discussing the effects of parental *attachment* on youth in the Arab community is essential. Fathers and mothers have different roles within the family and in nurturing their children [61]. The interviewed boys did not find their fathers to be supportive and helpful for a variety of reasons. First, the fathers were not often present at home. A study that examined family drawings of Bedouin-Arab children of polygamous families in Israel revealed that 36% of the children did not draw their father at all [62]. Second, the fathers are expected to take responsible for working and supporting the family, while women are expected to take care of the children's needs, including their education [63]. The interviewees reported that it was almost impossible for them to turn to their fathers for support and to share their difficulties with them, as this would be seen as a weakness. Third, some of the boys described extreme-risk family situations, where the fathers were violent and abusive toward them and their mothers.

However, while the boys expressed respect for their mothers and turned to them for support, they did not feel that their mothers could help them when they got into trouble with friends, at school, or with the police. Because women in Arab society are expected to

be obedient and dependent, the mothers' ability to help their children is generally less than that of the fathers [11,64].

Regarding the impact of the socioeconomic status of Arab society, in families with socioeconomic hardships and those that lack positive parental figures, attachment can be achieved through positive relationships with other significant people that the youth encounter in school or at social agencies. They generally described the inability of the schools and teachers to play a significant role in their lives. This is perhaps not surprising given the poor economic and professional environments in which Israeli Arab schools operate [5,65]. At the same time, however, most of the participants mentioned one or two teachers to whom they felt attached. They felt that these teachers helped them to not become involved in delinquency, because the youth did not want to disappoint them. They mentioned their positive treatment and their concern, which sometimes differed dramatically from the way they were treated by other teachers and by their parents. These teachers also served as a sympathetic ear and a source of support when needed. Unfortunately, there were very few of these positive figures in the participants' lives.

Some of the participants expressed positive attitudes toward their probation officers. They described them in a positive manner and greatly appreciated their work with them. They talked about them as central figures to whom they felt bonded and attached, and because of their relationship they felt they had succeeded in making positive changes in their lives.

Social bond theory's explanation of peer effects ignores the cultural context of traditional societies such as the Arab community in Israel [35,36]. The boys described strong relationships with peers. They felt strongly protective of their friends, even to the point of fighting on their behalf. A pronounced sense of traditional masculinity is a central element in the socialization of males in Arab culture. Males are traditionally expected to express toughness and strength [39,40,66]. Furthermore, in cultures where youth feel threatened and oppressed by external forces, relationships with peers might not prevent delinquency but, on the contrary, might be seen to increase it. Youth in Arab communities can be strongly affected by masculine honor culture codes that expect them to protect their friends, even if this requires being involved in violence and antisocial acts [67].

The *commitment* component of social bond theory can be better applied cross-culturally if it expands its focus from the individual's characteristics to a wider perception that includes both individual and contextual factors that obstruct children and youth from committing to school. In our case, the combination of scarce resources allocated to Israeli Arab schools, together with the lack of teachers' ability to deal with many of the boys, highlights the inability of the school to provide youth with the help they need to deal with their individual difficulties.

Arar and Mustafa [68] indicate that the financial resources and management services provided to Arabs in Israel are much lower than those available to Jews. It has also been found that more Arab children and adolescents are under-diagnosed with or untreated for ADHD than are their Jewish counterparts [69]. Mahajnah et al. [70] indicate that this may be because of the Arab population's poor access to services. Diagnosis can be expensive, as it is not covered by healthcare insurance in Israel; thus, poor families may not have sufficient resources to cover it.

According to Hirschi's [22] theory, lack of *involvement* in recreational activities increases the risk of involvement in delinquent behavior. Most of the youth said they spent a lot of time wandering around the neighborhood because they were bored and, as a result, their involvement in violent and antisocial behavior increased. A look at the allocation of resources for recreational activities in Arab neighborhoods in Israel reveals systematic discrimination which impairs the development of activities and social services. In 2008, mapping conducted in the northern part of Israel found that an estimated 5% of Arab children and youth participated in informal education, versus 30% of Jewish children and youth [71].

Another study of youth in Israel [60] found that Arab youth participated less than Jewish youth in community center activities, youth movements, and extracurricular activities. Almost 50% of the youth who participated in their study felt they had nowhere to spend their leisure time. One explanation for the low participation rate of Arab youth in informal educational activities is the insufficient allocation of budget and resources, which results in the inadequate supply and obstructed functioning of social services in the Arab communities [60].

The latter explanation is relevant to the last component of Hirschi's [22] theory: *belief.* The boys felt distrust towards services offered by the state, which are in part responsible for maintaining societal norms and rules. The fact that they lacked faith in these institutions contributed to their lack of involvement. According to Hasisi and Weisburd [47], Arabs believed that the police's response time to calls made by Arab citizens reporting on severe crimes in their community was much longer than average. Therefore, in order to better understand youth's lack of involvement, social bond theory should expand to discuss contextual factors related to family, social services offered by the state, and neighborhood socioeconomic conditions.

### 18. Limitation and Future Directions

Despite its contribution to theoretical and empirical literature, the current study has several limitations. First, it is based on a small sample (15) of Arab boys. Future studies should use a larger number of participants, including female adolescents, which will allow examining the cross-gender applicability of Hirschi's [22] theory. Second, all the data were based on youth informants; adding other informants such as parents and teachers many enrich the findings. Third, the interviews were translated into Hebrew, and only then into English. Still, the fact that some of the researchers were Arabic speakers allowed us to cross-reference with the original interview conducted in Arabic.

### 19. Conclusions

The study's findings reinforce the uniqueness of the situation in which Israeli Arab adolescents live. It is a complex situation that reflects their economic, social, and political status, and that seems to have an impact on their involvement in antisocial behavior and delinquent activity. Due to the structural constraints created by poverty, inequality, and oppression the youth in our study had limited opportunities to develop positive attachments, commitment, and involvement, which are important for preventing engagement in criminal behavior. These findings can be used to enrich the discussion of Hirschi's theory. Another important contribution of this study is to expand the literature of the relativity of Hirschi's theory on the Arab adolescent, which emphasis the important of considering these risk and protective factor in intervention and prevention programs dealing with adolescents, and minority adolescents in particular. We recommend that professionals and social workers encourage parents to be more involved in their adolescents' lives, especially fathers, and promote parent–child communication skills, which may decrease the tendency of adolescents to perpetrate aggression and violence.

**Author Contributions:** Conceptualization, M.K.-K., E.B.-C., M.A. and L.J.J.-L.; methodology, E.B.-C., M.A. and L.J.J.-L.; software, M.K.-K., E.B.-C., M.A. and L.J.J.-L.; validation, M.K.-K., E.B.-C., M.A. and L.J.J.-L.; formal analysis, M.K.-K., E.B.-C., M.A. and L.J.J.-L.; investigation, M.K.-K.; resources, M.K.-K., E.B.-C., M.A. and L.J.J.-L.; data curation, M.K.-K., E.B.-C., M.A. and L.J.J.-L.; writing—original draft preparation, E.B.-C., M.A. and L.J.J.-L.; writing—review and editing, M.K.-K., E.B.-C., M.A. and L.J.J.-L.; visualization, M.K.-K., E.B.-C., M.A. and L.J.J.-L.; supervision, M.K.-K., E.B.-C., M.A. and L.J.J.-L.; project administration, M.K.-K., E.B.-C., M.A. and L.J.J.-L. All authors have read and agreed to the published version of the manuscript.

**Funding:** This research received no external funding.

**Institutional Review Board Statement:** The study was approved by the Ethic Committee, Paul Baerwald School of Social Work and Social Welfare, The Hebrew University of Jerusalem.(protocol code 24032013 and date of approval: 24.03.2013).

**Informed Consent Statement:** All participants gave their informed consent to participate in the study.

**Data Availability Statement:** Since the research is confidential and anonymous, and anonymity was guaranteed to the participants, the data is kept in the department where the research was carried out.

**Conflicts of Interest:** The authors declare no conflict of interest.

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
