# Peer review of "Arab Youth Involvement in Delinquent Behaviors: Exploring Hirschi’s Social Bond Theory from a Qualitative Perspective"

_societies, doi:10.3390/soc13050128_

Round 1

Reviewer 1 Report

General Comments

This is a well-written manuscript with a good theoretical framework. As the authors acknowledged in the manuscript, the social bond theory is widely tested in the Western world, but not in other regions including the Arab world. Since this manuscript discusses how and why cultural differences might influence the way of social bond functions in the Arab world, this manuscript adds to the literature.

Yet, in my view, this manuscript can be improved by addressing the following points. First, why did the authors take a qualitative approach? Normally, researchers use a quantitative approach to test a theory. The authors should provide a justification for their methodological approach. Second, how is social bond theory utilised in qualitative research? Is it merely used as a theoretical framework? Or are the interview questions developed based on social bond theory? Since there is no example of the interview questions in the manuscript, it is unclear. Third, the authors should show the number of participants who mentioned one of the social bonds in the interview where relevant Show numbers where relevant (e.g. Line 320, Page 7).

It was my pleasure reading this manuscript before publication.

Specific Comments

There are two periods in Line 224, Page 5.

An indent is needed for a long quote (e.g. Lines 259-264, Page 6).

Author Response

We would like to thank the reviewers for their comments. In what follows, the comments appear in bold.

Please find the attached updated manuscript. 

Reviewers’ comments to Author:

Reviewer #1:
 Comments to the Author
General Comments

This is a well-written manuscript with a good theoretical framework. As the authors acknowledged in the manuscript, the social bond theory is widely tested in the Western world, but not in other regions including the Arab world. Since this manuscript discusses how and why cultural differences might influence the way of social bond functions in the Arab world, this manuscript adds to the literature.

Yet, in my view, this manuscript can be improved by addressing the following points. First, why did the authors take a qualitative approach? Normally, researchers use a quantitative approach to test a theory. The authors should provide a justification for their methodological approach. Second, how is social bond theory utilised in qualitative research? Is it merely used as a theoretical framework? Or are the interview questions developed based on social bond theory? Since there is no example of the interview questions in the manuscript, it is unclear. Third, the authors should show the number of participants who mentioned one of the social bonds in the interview where relevant Show numbers where relevant (e.g. Line 320, Page 7).

It was my pleasure reading this manuscript before publication.

 Thank you!

Specific Comments

There are two periods in Line 224, Page 5.

Corrected. Thank you.

An indent is needed for a long quote (e.g. Lines 259-264, Page 6).

This has been corrected throughout.

Reviewer 2 Report

See attached

Author Response

We would like to thank the reviewers for their comments. In what follows, the comments appear in bold.

Please find the Attached updated manuscript. 

Reviewer #2:
Comments to the Author
This is a much-needed study to expand the examination of criminological theory beyond Western lenses. That said, the paper is not sufficiently developed to warrant publication. While there are more specifics that follow, large problems consist of the following: (a) inadequate review of and rooting of paper in the literature; (b) research method described is not the method that is used for the findings; and (c) lack of mining of the interviews for new information that can add to the literature already in existence. It is likely that wholesale re-analysis of the data would reveal some original information that could substantially add to the paper for a publishable work.

We would like to thank the Reviewer for his/ her comments. We have made the changes suggested, by rooting the paper in the literature and in aligning the description of the methods with that used for the findings.   

Abstract

While references are generally not used in abstracts, if used, they should be those best suited to illustrate the points made. For example, Line 11- Use updated cross-cultural dimensions of social control theory (suggestions below).

We would like to thank the reviewers for his/her comments and suggestions. We have added an updated study related to social bond theory.

Line 16 - The paper doesn’t examine interventions; it examines theory from a cultural perspective.

We agree with the reviewer and have corrected the text accordingly.

Introduction

The introduction is absent. Discussion of delinquency and social bonds is left out before delving into Arab-Palestinian structure in Israel. Part of the last paragraph (approximately Line 61) would be a good start to an introduction.

We would like to thank the reviewer for this comment. We now start the Introduction with the paragraph suggested to emphasize the difference between Jewish and Arab society in the extent of violence.

Line 45 – Describe specific changes that have occurred in the family to better orient the reader; similar problem on Line 47.

Following the Reviewer's comment, we have clarified the sentence.

Theory/Literature Review

A transition is needed from the introduction to theory and literature review. Explain why social bonds are a fruitful avenue for the investigation of delinquency, but how there is a gap with non-western cultures. Authors should draw a link from the demographic information discussed to theory.

In general, there is a need to root this study in the recent scholarship on social bonds and delinquency (See Costello & Laub’s 2020 review of social control theory in Annual Review of Criminology) as well as the cross-cultural literature on delinquency and social control (ex: Cabrera, 2019; Junger & Marshall, 1997).

Following the Reviewer's comment, we have made the changes suggested.

Attachment

Line 82 – How is parental attachment defined here? Does it vary by cultural context?

Following the Reviewer's comment, we have clarified this paragraph.

Line 89 – This paragraph needs citations.

Following the Reviewer’s comment, we have added citations to this paragraph.

Line 101 – The key point is prosocial peers. That is not discussed here.

We have made the clarification as suggested.

Line 108 – This seems simplistic as modeling peer delinquency takes into account other factors.

Following the Reviewer’s comment, we have clarified this paragraph.

Line 116 – This is an important point, and it would be helpful to acknowledge that this point is made clearer in other theories and evidence (ex: “Street Code” work in US and South Africa in particular)

Following the reviewer's comment, we have added an example for a longitudinal study focusing on street code and attachment among youth as related to their involvement in violence (Page 6).

Commitment – more clearly articulate the gap between commitment and conventional lifestyle and use current social bond literature to support that gap exists.

Line 123 – Commitment to success or commitment to prosocial individuals, institutions, and activities?

Thank you for your comment. We now clarify that according to the theory, both aspects are relevant: commitment to succeed and involvement in conventional activities. 

Involvement

Line 150 – Unclear how unstructured leisure time is connected to lack of community services. It should be the opposite – better community services leads to more structured time, leaving less time to engage in delinquent activity.

We agree with the Reviewer’s comment and this is the direction we report on. We have expanded this part to make this point clearer.

Research Questions – both of these questions need some refinement.

There seems to be a strength and direction component of social bonds missing in Research Question 1. It seems to be a lack of social bonds, not existence of social bonds.

Thank you for drawing our attention to the problem in the research questions. We have made revised them as suggested by Reviewer #3.

Research Question 2 is unclear. Does “they” refer to social bonds in general or lack of prosocial bonds?

We would like to thank the reviewer for his/her comment. We now clarify this in both research questions.

Methods

Line 195 – approved by the interviewer?

Thank you. This has been corrected.

Were interviews electronically recorded or handwritten? Why were they transcribed into Hebrew?

The interviews were recorded with the agreement of the participants. As all participants are Arab, all interviews were conducted in Arabic. However, not all the researchers speak Arabic, therefore the interviews were translated into Hebrew.

Line 199 – “were” is missing.

Thank you. Corrected.

-More information about data analysis is needed. Were there any discrepancies in coding? If so, how were those resolved? What were the units of meaning? Was this axial coding? If so, say more. Given that the finds are oriented around the four social bonds, it seems like the method (grounded theory) was not used as this is an inductive method to come up with original themes to create theory. Instead, this analysis seems to be deductive relying on social bonds elements identified by Hirschi in 1969.

Following the Reviewer’s comment, we have added information in the analysis section and clarified that there were no discrepancies in coding. Units of meaning were anchored around the four elements of social bond theory: attachment, commitment, involvement, and belief.

The analysis of the data was deductive,  which means that we applied the theory to the data to test the theory.  The codes were based on concepts drawn from Hirschi's theory (Bingham & Witkowsky, 2022).

-Limitations should be covered in the Discussion section.

We have expanded and revised the Discussion to include a Limitations subsection.

-The demographic information about the youth (ex: age, family structure, etc.) is missing.

We would like to thank the reviewer for his/her comment. Following this comment we have added information regarding the participants’ demographics.

Findings

The findings are confusing when references to previous studies are included. Some of these are appropriate for the literature review section and some should be part of the discussion.

Thank you for your comment. We have made the changes suggested in the Findings and Discussion. 

There seems to be only a broad-brush general overview of how youth connect to family and school through social bonds. Can these youth be compared and contrasted? Are their deviant situations/cases that differ from the majority? It seems as though the contextualization of social bonds could be completed without the interviews. The interviews need to be mined further to enhance the points made by previous work and perhaps discover new ones.

In the paper we argue that the attachment of the adolescents to their mother is stronger than to their father. Also, they reported that the school system did not believe in and support them. Nevertheless, some reported positive attachment to their teachers and probation officers. We would have expected that the latter would report lower violence. But this is not the case: the findings indicate that the effect of social context is beyond the effect of other elements of social bond theory.

Discussion

This section should be rooted in the cross-cultural literature on social bond theory. Again, that literature is largely absent in this paper.

Following the Reviewer’s comment we have made changes in the literature review and rooted it with the theory.

Line 394 – What does “cross-sectional understanding mean? Should this be cross-cultural?

We have corrected the term to cross-cultural.

The discussion section ends abruptly. A paragraph summarizing the contributions of the paper to the literature may help.

Following the Reviewer’s comment, we have added a paragraph that summarizes the paper’s contributions.

Reviewer 3 Report

General and overall statement

·      This manuscript has potential and could make a contribution to the research. I agree with the researchers’ statement about the importance of the research to fill the gap in the literature regarding the use of social bond theory to explain youth delinquency in non-western cultures. However, I do have suggestions for improvement.

·      There are some minor editing concerns in regard to grammar and punctuation throughout the document. For example on page 5 there are two periods at the end of the first sentence of the findings subheading on line 224.

·      Throughout the document there are areas in which the font size changes, for example the top of page 4 and page 8 on line 362-363. Consistency with font size is needed.

Introduction, conceptual argument, literature review, and framing

·      More of a discussion is needed of the discrimination Israeli Arab youth experience. Are they policed more than Israeli Jewish youth? How does this contribute to their delinquency and the inequality in the rates of youth delinquency present in Israel? For example, one area where more discussion could be included of this discrimination towards Israeli Arab youth is on page 3 in the section discussing commitment. In the last paragraph of the Commitment section, the author(s) very briefly discuss inequality in the Israeli Education system. Much more discussion of the inequalities in the education system are needed here.

Data, Methods, Analytical Strategy, and Results

·      Also I recommend revising the research questions on page 4, to read:

o   To what extent are the components of Hirschi’s theory manifested among Israeli Arab delinquent youth?

o   To what extent are they influenced by the values and norms of the traditional society to which they belong and by the social, political, and economic disadvantages experienced by the Arab community in Israel?

·      More detailed description of the data analysis is needed. For example, explain what the researcher(s) mean by becoming “empathically acquainted with the interviewees’ narratives” in the first stage of analysis. Also, providing example of the “units of meaning” or the themes that emerged in their analysis would strengthen the data analysis section.

·      The citation of the participant’s quotes are a bit unclear. For example on page 5 the author(s) states that “U. describes his relationship with his father” and then proceeds with a quote that at the end of the quote has the citation “ (Hussain).” Is this quote from U. or from Hussain? A clearer indication of which participant is quoted would be helpful to readers to be able to follow the participants narratives and the findings.  

·      In the discussion of the findings, especially on page 6, the author(s) touch on traditional gender roles in Arabic culture. However, more discussion and unpacking is needed of the impact of these gender roles and ideologies of masculinity on youth, especially the young men in this study and how it is connected to and/or impacts their behavior and/or delinquency.

Author Response

We would like to thank the reviewers for their comments. In what follows, the comments appear in bold.

Please find the attached updated manuscript. 

Reviewer #3:

Comments and Suggestions for Authors

General and overall statement

  • This manuscript has potential and could make a contribution to the research. I agree with the researchers’ statement about the importance of the research to fill the gap in the literature regarding the use of social bond theory to explain youth delinquency in non-western cultures. However, I do have suggestions for improvement.
  • There are some minor editing concerns in regard to grammar and punctuation throughout the document. For example on page 5 there are two periods at the end of the first sentence of the findings subheading on line 224.

Corrected. Thank you (see also the comment of Reviewer #1).

  • Throughout the document there are areas in which the font size changes, for example the top of page 4 and page 8 on line 362-363. Consistency with font size is needed.

Corrected. Thank you.

Introduction, conceptual argument, literature review, and framing

  • More of a discussion is needed of the discrimination Israeli Arab youth experience. Are they policed more than Israeli Jewish youth? How does this contribute to their delinquency and the inequality in the rates of youth delinquency present in Israel? For example, one area where more discussion could be included of this discrimination towards Israeli Arab youth is on page 3 in the section discussing commitment. In the last paragraph of the Commitment section, the author(s) very briefly discuss inequality in the Israeli Education system. Much more discussion of the inequalities in the education system are needed here.

Thank you for your comments. We have now added information about the discrimination of Arab youth in the education system and its impact on involvement in violence.

Data, Methods, Analytical Strategy, and Results

  • Also I recommend revising the research questions on page 4, to read:

o   To what extent are the components of Hirschi’s theory manifested among Israeli Arab delinquent youth?

o   To what extent are they influenced by the values and norms of the traditional society to which they belong and by the social, political, and economic disadvantages experienced by the Arab community in Israel?

  • More detailed description of the data analysis is needed. For example, explain what the researcher(s) mean by becoming “empathically acquainted with the interviewees’ narratives” in the first stage of analysis. Also, providing example of the “units of meaning” or the themes that emerged in their analysis would strengthen the data analysis section.

Following the Reviewer’s comment, we have added information in the analysis section (see also comment to Reviewer #2).

  • The citation of the participant’s quotes are a bit unclear. For example on page 5 the author(s) states that “U. describes his relationship with his father” and then proceeds with a quote that at the end of the quote has the citation “ (Hussain).” Is this quote from U. or from Hussain? A clearer indication of which participant is quoted would be helpful to readers to be able to follow the participants narratives and the findings.

Thank you for your comment. We have made changes in the quotations to make this clear.

  • In the discussion of the findings, especially on page 6, the author(s) touch on traditional gender roles in Arabic culture. However, more discussion and unpacking is needed of the impact of these gender roles and ideologies of masculinity on youth, especially the young men in this study and how it is connected to and/or impacts their behavior and/or delinquency.

We have expanded the discussion section to also address traditional gender roles in Arab culture.

Thank you for your helpful comments.

Sincerely
